# Immunohistochemical Expression of Glutathione Peroxidase 1 (Gpx-1) as an Independent Prognostic Factor in Colon Adenocarcinoma Patients

**DOI:** 10.3390/ph16050740

**Published:** 2023-05-12

**Authors:** Marlena Brzozowa-Zasada, Adam Piecuch, Karolina Bajdak-Rusinek, Kamil Janelt, Marek Michalski, Olesya Klymenko, Natalia Matysiak

**Affiliations:** 1Department of Histology and Cell Pathology in Zabrze, Faculty of Medical Sciences in Zabrze, Medical University of Silesia in Katowice, 40-055 Katowice, Poland; 2Department of Medical Genetics, Faculty of Medical Sciences in Zabrze, Medical University of Silesia in Katowice, 40-055 Katowice, Poland

**Keywords:** glutathione peroxidases, colon adenocarcinoma, oxidative stress, antioxidant enzymes, immunohistochemistry, cancerogenesis

## Abstract

Several studies revealed that expression levels of glutathione peroxidase 1 (Gpx-1) can be associated with cancer development, mainly through its role in hydroperoxide scavenging by regulating intracellular reactive oxygen species (ROS) levels. Therefore, our aim was to investigate the expression of Gpx-1 protein in a population of Polish patients with colon adenocarcinoma in the absence of any therapy prior to radical surgery. The study was carried out using colon tissue from patients with adenocarcinoma of the colon confirmed by histopathological examination. Gpx-1 antibody was used to determine the immunohistochemical expression of Gpx-1. The Chi^2^test or Chi^2^_Yatesa_ test were used to analyse the associations between the immunohistochemical expression of Gpx-1 and clinical parameters. The relationship between Gpx-1 expression, and 5-year patient survival was examined using Kaplan–Meier analysis and the log-rank test. Intracellular localisation of Gpx-1 was detected by the use of transmission electron microscopy (TEM). Western blot analysis was used for the evaluation of Gpx-1 protein expression levels in cancer cell lines in vitro. Immunohistochemical study revealed that the high expression of Gpx-1 was associated with the tumour’s histological grade, proliferating cell nuclear antigen (PCNA) immunohistochemical expression, depth of invasion, and angioinvasion (all *p* < 0.001) (4). The high immunohistochemical expression of Gpx-1 is correlated with poor prognosis of colon adenocarcinoma patients.

## 1. Introduction

Colorectal cancer (CRC) is estimated to have caused 1.9 million new cases and 0.9 million deaths worldwide in 2020, making it the third most common malignancy and the second most deadly cancer [1]. Alcohol consumption, smoking, unhealthy diet, and obesity are important factors associated with the development of this type of cancer [2]. Among the factors that have a correlation with the development of the disease, it is also worth mentioning ageing, genetic mutations, and hereditary factors [2]. The most common subtype of CRC is colorectal adenocarcinoma (COAD), which develops from the epithelial cells as a sequence of specific morphological and genetic changes. Under pathological conditions, the shape of the epithelial cells changes and they grow out of control, leading to the development of adenoma and adenocarcinoma [3].

As screening for colorectal cancer has increased, the incidence of COAD has decreased significantly, but mortality is predicted to increase by 60% when comparing between 2013 and the projection for 2035 [4]. It should be noted that despite advances in currently available standard therapies, such as surgery, chemotherapy, radiotherapy, and immunotherapy, the 5-year OS of patients with COAD remains poor [4,5]. Thus, the development of a novel biomarker is urgently needed to improve patient’s outcome, allow earlier therapeutic intervention, and reduce the increasing burden [6,7].

Vertebrate glutathione peroxidases comprise eight members, GPx 1–8, which are involved in several important biological processes. Their main role is to act as a catalyst for the reduction of H_2_O_2_, which leads to the oxidation of glutathione (GSH). GSH can then be reduced back to nicotinamide adenine dinucleotide phosphate (NADPH) by GSH reductase [8]. Gpx-1, which is a highly expressed member of the GPX family and is present in all cells, is one of these enzymes. The location of the Gpx-1 gene is associated with chromosome 3p21.31. It has 2 exons and is 1178 base pairs long. It possesses five transcribed variants and it is transcribed in a negative direction. In mammals, the active Gpx-1 protein is a homotetramer. It consists of four identical subunits with a molecular weight of 22–23 kDa. There are approximately 208 amino acids in each monomer [9]. Being a selenoprotein, the selenium within the Gpx-1 protein is formed by the addition of the 21st amino acid, selenocysteine, into the nascent polypeptide chain during the translation process at the UGA stop codon [10]. The expression of Gpx-1 is mainly regulated at the transcriptional level by a number of transcription factors and by oxygen tension. Moreover, the expression of Gpx-1 is controlled post-transcriptionally by the availability of selenium and cofactors required for Sec production and insertion. The localisation of Gpx-1 is associated with the cytoplasm, mitochondria, nucleus, and peroxisomes [8,9,10,11]. Different locations of GPx-1 may correspond to different functions, i.e., GPx-1 in the cytosol can remove reactive peroxides, such as H_2_O_2_ [12], while GPx-1 in the mitochondria can prevent oxidative damage to mitochondrial DNA [13].

Several studies revealed that high or low levels of Gpx-1 expression can be associated with cancer development, mainly through its role in hydroperoxide scavenging by regulating intracellular ROS [8,9,10]. In this place, it should also be noted a that Gpx-1 Pro198Leu polymorphism has been extensively studied in breast [14], leukaemia [15], and colorectal cancer [16]. Interestingly, in the European population, no associations between the Gpx-1 Pro198Leu polymorphism and colorectal cancer risk were observed in any of the cases studied [16,17]. However, data on the clinical application of Gpx-1 expression in European colorectal cancer patients, especially colorectal adenocarcinoma, are scarce. Against this background, we investigated the Gpx-1 protein expression in a population of European (Polish) patients with colon adenocarcinoma in the absence of any therapy prior to radical surgery. The association between Gpx-1 protein expression and clinicopathological variables was also examined. Moreover, the expression of Gpx-1 was related to the expression of proliferating cell nuclear antigen (PCNA) which is commonly present in proliferating cells and tumour cells. PCNA is reported to be directly linked to the synthesis of cellular DNA, to the grade of tumour differentiation, and to tumour prognosis [18,19]. The prognostic activity of Gpx-1 protein was analysed regarding the 5-year survival of patients, which is a very important factor from a clinical oncological point of view. Moreover, by using the immunogold labelling method, the intracellular localisation of Gpx-1 was detected within the cells of cancer tissues. By use of the Western blot method, we assessed the expression of Gpx-1 in three distinct cancer cell lines and in a control cell line.

## 2. Results

### 2.1. Characteristics of Patients Included in the Study

A total of 72 men and 71 women were included in our study (mean age: 65 years; range: 56 to 77 years). In 68 (47.55%) cases, the cancers were located in the right part of the colon, and 75 (52.45%) were in the left part of the colon. They were classified according to the following 3 histological grades of differentiation: G1—34 cases (23.78%), G2—68 cases (47.55%), and G3—41 cases (28.67%) (Table 1). In samples of colon adenocarcinoma, the positive immunohistochemical reaction indicating the presence of Gpx-1 protein was observed in the cytoplasm and nuclei of both the cancer cells and stromal cells. In the cells of nonpathological colon tissue (Figure 1), a positive reaction was also detected. Importantly, expression was described as strong in the vast majority of colon adenocarcinoma tissues, whereas expression in cells of the adjacent non-pathological colon mucosa was found to be low.

### 2.2. The Associations between the Immunohistochemical Expression of Gpx-1 and Patients’ Clinical Features

Among the study cohort, 74 (51.75%) colon adenocarcinoma samples demonstrated the high immunohistochemical expression of Gpx-1 protein, whereas 69 (48.25%) showed low immunoreactivity. In the next step, results of the immunohistochemical analysis were correlated with clinicopathological features of patients and 5-year survival rates. Gpx-1 expression was significantly associated with tumour histological grade (*p* < 0.001, Chi^2^test). In 4 (11.76%), 35 (51.47%), and 35 (85.37%) of the G1, G2, and G3 tumours, respectively, high levels of Gpx-1 protein expression were found. The low level of immunohistochemical expression of Gpx-1 protein was found in 30 (88.24%), 33 (48.53%), and 6 (14.63%) of the G1, G2, and G3 tumours, respectively. In addition, Gpx-1 expression was associated with immunohistochemical expression of PCNA (*p* < 0.001 Chi^2^test). Gpx-1 protein was found to be highly expressed in 14 (17.95%) and 64 (82.05%) samples with low and high PCNA immunoreactivity, respectively (Table 2; Figure 2).

Notably, Gpx-1 expression was also associated with angioinvasion (*p* < 0.001, Chi^2^test). High Gpx-1 immunohistochemical expression was found in 69 (66.35%) of the patients with positive angioinvasion, whereas low immunoreactivity was found in 35 (33.65%). On the other hand, 5 (12.82%) patients without angioinvasion had high Gpx-1 expression, whereas 34 (87.18%) patients had low Gpx-1 immunoreactivity. Gpx-1 immunohistochemistry was also related to the depth of invasion (T) (*p* < 0.001, Chi^2^test). Among the patients who were characterised as T1, a high level of immunohistochemical reaction was observed in 3 patients (13.04%) and a low level of expression was detected in 20 patients (86.96%). For the T2 patients, a strong immunohistochemical reaction for Gpx-1 was reported in 10 (47.62%) patients, whereas a low level of expression was detected in 11 (52.38%) patients. In the T3 and T4 groups, high expression was noted in 44 (59.46%) and 17 (68%) patients, respectively (Table 3). In patients with stage I of the disease, 13 (6.98%) showed a high level of Gpx-1 expression and 21 (61.76%) demonstrated a low level of reaction. In patients with stage II of the disease, 23 (71.88%) demonstrated a high level of immunoreaction and 9 (28.13%) revealed a low level. In contrast, patients with stage III of the disease revealed a high and low level of immunoreactivity in 38 (49.355) and 39 (50.65%), respectively.

### 2.3. The Prognostic Significance of Gpx-1 Expression in Relation to 5-Year Survival

The prognostic significance of Gpx-1 expression in colon adenocarcinoma patients was analysed in relation to 5-year survival. All samples were evaluated with Kaplan-Meier survival curves. The 5-year survival rate was significantly higher in the group with low Gpx-1 expression (log-rank, *p* < 0.001) (Figure 3).

In addition, the value of Gpx-1 expression in the context of 5-year survival was evaluated in subgroups of patients stratified according to the grade of histological differentiation, the depth of invasion, the staging, and the expression of PCNA (Figure 4). Notably, Gpx-1 expression was not associated with 5-year survival in patients stratified by G1 (log-rank test, *p* = 0.412, G2 (log-rank test, *p* = 0.181), and G3 (log-rank test, *p* = 0.007). However, patients with low Gpx-1 immunohistochemistry had significantly longer 5-year survival compared to patients with high Gpx-1 immunohistochemistry (log-rank test, *p* < 0.011). Similar results were obtained in patients with a T3/T4 depth of invasion (log-rank test, *p* = 0.001). Furthermore, in patients with stage I disease, low Gpx-1 expression was associated with 5-year survival (log-rank test, *p* < 0.001). Low Gpx-1 expression was also associated with 5-year survival in patients with stage III disease. However, in this case, the results were not statistically significant (log-rank test, *p* = 0.052). The low expression of Gpx-1 was associated with better 5-year survival rate both in patients with high and low PCNA immunohistochemical expression (log-rank test, *p* = 0.001).

Gpx-1 immunohistochemical expression, histological differentiation grade, invasion depth, angioinvasion, and PCNA expression were significant prognostic factors in univariate Cox regression analyses. In our cohort of patients, the grade of tumour differentiation and Gpx-1 expression were found to be independent prognostic factors for 5-year survival in patients with colon adenocarcinoma (Table 4).

### 2.4. Detection of Gpx-1 at the Cellular Level by the Use of TEM

The localisation of the Gpx-1 protein at the cellular level in colorectal adenocarcinoma samples was demonstrated using an immunogold labelling method. In cancer cells, black granules indicating the presence of Gpx-1 were detected in the cytoplasm. In the mitochondria and cisterns of the rough endoplasmic reticulum, electron-dense granules were also found. Gpx-1 was also detected in the plasma membrane and in the cytoplasm in the vicinity of the plasma membrane. In colonocytes from non-pathological samples, scattered black granules were found in the cytoplasm of the apical part of cells (Figure 5).

### 2.5. Analysis of Gpx-1 in Selected Cancer Cell Lines by the Use of the Western Blot Method

The Western blot technique was used to assess protein expression levels. Using this method, the Gpx-1 protein level was determined in cancer cell lines in vitro. To investigate differences in protein expression in cell lines that may represent different types of colorectal cancer, the SW1116 (Duke A), LS 174T (Duke B), and HCA-2 (Duke C) cell lines were selected for the study. The CCD 841 CoN line was used as a control group.

The study showed that the highest level of Gpx-1 protein expression was found in the HCA-2 cell line, which represents the Duke C stage of the disease (stage III). The lowest level of expression was observed in the SW1116 cell line, which represents the Duke A stage of the disease (stage I). Statistically significant differences in Gpx1 protein expression were found between HCA-2 cells and SW1116 cells, between CCD 841CoN cells and SW1116 cells, and between LS 174T cells and SW1116 cells (Figure 6).

## 3. Discussion

Gpx-1 is an antioxidant enzyme that plays a pivotal role in protecting cells from the effects of oxidative stress. The increased expression of this protein in cancer cells suggests that Gpx-1 may play an important role in carcinogenesis and disease progression [8]. Studies have shown that Gpx-1 can regulate cancer cell proliferation, invasion, migration, apoptosis, immune response, and drug sensitivity. It has also been shown to be a promising prognostic biomarker and has a strong association with tumour clinicopathological features [20].

It should be noted that our study is the first to evaluate the clinical application of the expression of Gpx-1, in particular in relation to the 5-year survival rate in patients with colon adenocarcinoma from Europe. In our cohort of patients, approximately 78% of the colon adenocarcinoma specimens showed a high level of Gpx-1 protein expression, while a low level of immunoreactivity was found in only 22% of the cases. Gpx-1 expression was detected within the cytoplasm. TEM studies using the immunogold labelling method confirmed that Gpx-1 in colon adenocarcinoma cells showed cytoplasmic expression. Within the cytoplasm, the expression was evident in the cell membrane, in the endoplasmic reticulum membranes, and in the mitochondria. The localisation of GPx-1 within cells or within cell organelles, in particular the mitochondria, may be associated with different functions performed by Gpx-1. For example, Gpx-1 in the cytosol is associated with the removal of peroxides [12], whereas GPx-1 in the mitochondria can prevent oxidative damage to mitochondrial DNA [13]. Statistical analysis revealed that the high immunohistochemical expression of Gpx-1 was significantly correlated with tumour histological grade (*p* < 0.001, Chi^2^test), depth of invasion (*p* < 0.001, Chi^2^test), angioinvasion (*p* < 0.001, Chi^2^test), and PCNA immunohistochemical expression (*p* < 0.001, Chi^2^test). It is interesting to note that high levels of Gpx-1 protein expression were found in only 12% of the G1 tumours, 51% of the G2 tumours, and 85% of the G3 tumours. In patients with stage I disease, 38% showed high levels of Gpx-1 expression, in patients with stage II disease, 72% exhibited high immunohistochemical expression of Gpx-1, and in patients with stage III disease, approximately 50% showed high expression of this protein. Furthermore, by analysing the results of Gpx-1 protein expression in different cell lines representing Duke stages A, B, and C, which correspond to stages I, II and III of this malignancy, respectively, it can be concluded that these results do not differ from the data obtained by analysing patient tissues, where the highest level of expression was found in patients with stage II and III of the disease and the lowest was found in patients with stage I of the disease. These results may indicate that Gpx-1 plays an important role in the progression of colorectal adenocarcinoma. It could be used as a potential biomarker to select patients with a more aggressive form of this malignant tumour. In this context, it is worth noting that the expression of Gpx-1 was also associated with the immunohistochemical expression of PCNA (*p* < 0.001, Chi^2^_Yatesa_ test). In patients with low levels of PCNA expression, 85% revealed the low level of Gpx-1 expression and 15% demonstrated the high level of Gpx-1. In contrast, in patients with high expression of PCNA, only 18% revealed the low level of Gpx-1 expression, and 82% demonstrated the high level of Gpx-1 expression. There was also a significant statistical difference in survival between the patients in the T1/T2 group. The patients with a high level of Gpx-1 expression had a significantly shorter survival time compared to the group with a low level of Gpx-1 expression (log-rank, *p* < 0.001). There were similar results in the T3/T4 group (log-rank, *p* = 0.010).

As mentioned earlier, Gpx-1 is closely associated with the development of tumours and with the survival and prognosis of patients in a wide range of human malignant tumours [21]. Our findings are similar to those of several reports. In acute myeloid leukaemia patients, high Gpx-1 expression was associated with poor prognosis [22]. Similar results have been obtained in patients with renal cell carcinoma (RCC), where high expression of Gpx-1 is positively associated with distant metastases, lymph node metastases, and tumour stage [23]. In oral squamous cell carcinoma [24], laryngeal squamous cell carcinoma [25], and malignant pleural mesothelioma [26], increased expression is also clearly associated with patient prognosis and distant metastasis. In contrast, in pancreatic cancer patients, the expression of Gpx-1 is decreased. This low level of expression predicts a poor prognosis. Meng et al. demonstrated that in a pancreatic cancer cell line, silencing of Gpx-1 promotes a mesenchymal phenotype and gemcitabine resistance through activation of the ROS-mediated Akt/GSK3β/Snail signalling axis [27]. Furthermore, in pancreatic ductal adenocarcinoma cells, activation of ROS/AMPK signalling and Gpx-1 degradation may promote the induction of protective autophagy to survive in a glucose-starved tumour microenvironment. Both high expression of Gpx-1 and suppression of autophagy may sensitise cells to starvation-induced cell death by activating apoptotic cell death, which is linked to caspase activity [28].

In conclusion, Gpx-1 was identified as a protein associated with decreased 5-year survival in patients with colon adenocarcinoma based on the results of the Cox regression model. We have found that the degree of histological differentiation and the immunohistochemical expression of Gpx-1 were independent prognostic factors for adenocarcinoma. Notably, our study is the first to demonstrate immunohistochemical expression of Gpx-1 in colon adenocarcinoma tissue in patients from European populations. Furthermore, it demonstrates the prognostic value of Gpx-1 expression in patients stratified according to certain clinical oncologically relevant criteria. In this case, tumour grade, depth of invasion, PCNA expression, and staging were considered. Our work is also the first to show the localisation of Gpx-1 in the tumour tissue at the level of the electron microscope, using the immunogold labelling method. However, there are some limitations to our study. These are as follows. The sample size of the study was small and the patients were from one hospital, possibly introducing selection bias. In future studies, the sample size should be increased. In addition, studies with PCR techniques—especially in patients—would be advisable after the collection of study material from a larger number of patients. In addition, a silencing or gene transfer study should be performed in colorectal cancer cell lines.

## 4. Materials and Methods

### 4.1. Patients and Tumour Samples

For the study, tissue from the colon was obtained from patients undergoing resection of the colon with histopathologically confirmed colon adenocarcinoma at the Jaworzno Municipal Hospital between January 2014 and December 2015. The following patients were excluded from the study: those treated with preoperative radiotherapy or chemotherapy, patients with serious medical complications or distant metastases, patients being resected due to tumour recurrence, patients with adenocarcinomas related to inflammatory bowel disease, and patients whose histopathology revealed a subtype other than adenocarcinoma. According to an established protocol, histopathological sections were taken from each surgical specimen, including tumour fragments and sections of adjacent tissue free of tumour changes. The samples were fixed in formalin and embedded in paraffin blocks. Paraffin blocks were then cut and sections were routinely stained with H&E to establish the histopathological diagnosis. Tissue margin sections were also analysed. If any cancerous cells were present, then the material was not included in the study. To assess the prognostic importance of Gpx-1 protein, we followed the patients for a period of 5 years to evaluate the 5-year survival rate.

### 4.2. Immunohistochemical Staining

Paraffin-embedded tissue blocks containing colon adenocarcinoma samples fixed in formalin were cut into 4 m thick sections, fixed on polysine slides, deparaffinised in xylene, and rehydrated through a graded series of alcohol. For retrieval of antigenicity, sections were microwave-treated. Subsequently, the sections were incubated with antibodies against Gpx-1 (polyclonal antibody from GeneTex. Cat. No. GTX03346, final dilution 1:600, Irvine, CA, USA) and PCNA (GeneTex. polyclonal antibody. Cat. No. GTX100539, final dilution 1:600, Irvine, CA, USA). For visualisation of protein expression, the sections were treated with BrightVision (Cat. No. DPVB55HRP WellMed BV, ’t Holland 31, 6921 GX Duiven, The Netherlands) detected system and Permanent AP Red Chromogen (Dako LPR from Agilent Technologies Code K0640), and Mayer’s haematoxylin was used as a nuclei counterstain. Furthermore, healthy mucosa sections from patients undergoing a screening colonoscopy free of inflammatory or cancerous lesions were analysed for the expression of Gpx-1 and PCNA. For the analysis of the results of the immunohistochemical staining, we have adapted the immunoreactive score on the basis of previous publications [29,30]. The intensity was scored in the following way: 0, no signal; 1, weak; 2, moderate; 3, strong staining. The frequency of positive cells was determined semi-quantitatively by assessing the whole section and each sample was scored on a scale of 0 to 4: 0, negative; 1, positive staining in 10–25% cells; 2, staining in 26–50% cells; 3, staining in 51–75% cells; and 4, staining in 76–100% cells. Finally, a total score from 0 to 12 was calculated and graded as follows: I, score 0 to 1; II, score 2 to 4; III, score 5 to 8; IV, score 9 to 12. Grade I was considered negative and grades II, III, and IV were considered positive. Grades I and II represented no or weak staining (low expression) and grades III and IV represented strong staining (high expression), respectively.

The evaluation was carried out by two independent pathologists. Differences were again assessed until a consensus was obtained.

### 4.3. Statistical Analysis

Statistica 9.1 software (StatSoft, Cracow, Poland) was used to analyse the correlation between immunohistochemical expression of Gpx-1 and clinical parameters. Medians and ranges were used for all quantitative variables. To compare the analysed groups, the Chi2 and Chi2- Yates tests were used. The relationship between the intensity of Gpx-1 expression and 5-year survival of patients was tested using Kaplan–Meier analysis and log-rank tests. The results were considered to be statistically significant if *p* < 0.05.

### 4.4. Immunogold Electron Microscopy

For the study using immunogold labelling procedures, samples were fixed in 4% paraformaldehyde in 0.1 M phosphate-buffered saline (PBS) at room temperature for 2 h and subsequently washed several times in PBS. Following washing, the specimens were dehydrated in a graded ethanol series and infiltrated in a 2:1 (*v*:*v*) ethanol/LR White mixture and 1:2 (*v*:*v*) for 30 min each on ice. The samples were then infiltrated in pure LR White. Ultrathin sections (70 nm) were cut using an RMC Boeckeler Power Tomo PC ultramicrotome with a diamond blade (45°; Diatom AG, Biel, Switzerland). Ultrasections were mounted on Formvar-coated 200 mesh nickel grids and immunolabelled. Sections on the grids were first pre-incubated for 30 min by floating on drops of 50 mM NH4 Cl in PBS and then blocked for 30 min on drops of 1% BSA in PBS. The grids were then incubated overnight (16–18 h) at 4 °C with primary anti-Gpx-2 antibody diluted 1:20 in BSA. The bound antibodies were localised by incubating the sections for 1 h on immunogold conjugated goat anti-mouse IgG 15 nm (BBInternational BBI Solutions, Sittingbourne, UK) diluted 1:100. Finally, the grids were washed on PBS drops (five changes, 5 min each) and water (three changes, 3 min each) before staining with 0.5% aqueous uranyl acetate. Controls did not use the primary antibody. The grids were then air dried and analysed in a TECNAI 12 G2 Spirit Bio Twin FEI Company transmission electron microscope at 120 kV. Images were captured using a Morada CCD camera (Gatan RIO 9, Pleasanton, CA, USA).

### 4.5. Cell Lines

Three colorectal cancer cell lines were used in the experiments: HCA-2 (Duke C), LS 174T (Duke B), and SW1116 (Duke A), together with a normal cell line, CCD 841 CoN. All cell lines were supplied by ATCC (American Type Culture Collection ATCC^®^, Old Town Manassas, VA, USA). To ensure optimal conditions for cell growth, appropriate culture media were used: Eagle’s minimum essential medium (EMEM) (ATCC 30-2003) for CCD 841CoN and LS 174T cell lines, and Dulbecco’s modified Eagle’s medium/Nutrient Mixture F-12 Ham (DMEM), Sigma-Aldrich D8437). For SW1116 and HCA-2 cell lines, both media were supplemented with 10% foetal bovine serum ((FBS), ATCC 30-2020) and 1% penicillin–streptomycin–neomycin stabilised solution (Sigma-Aldrich P4083).

### 4.6. Western Blot

Whole-cell lysates were obtained from the colorectal cancer cell lines (HCA-2, LS 174T, SW1116) and the control cell line (CCD 841 CoN) by the use of PierceTM lysis buffer (Cat. numb. 87,788; Thermo Fisher Scientific, Waltham, MA USA) with the addition of Halt™ Protease Inhibitor Cocktail 100x (Thermo Fisher Scientific) and 0.5 M EDTA solution. Here, 8 µg of protein were separated by 15% sodium dodecyl sulphate-polyacrylamide gel electrophoresis (SDS-PAGE) and transferred to nitrocellulose membranes for 2 h using the tetra cell-blot (Biorad, Müchen, Germany) with 1x blotting buffer (20 mM Tris, 150 mM glycine, 20% methanol, pH 8.3). Western blot protein detection was performed with rabbit anti-glutathione peroxidase 1 (GeneTex, EPR3312; 1:1000) and mouse anti-β-Actin (R&D SYSTEM, MAB8929; 1:5000) antibodies in 5% dry milk/TBS/Tween, followed by species-specific secondary HRP-coupled antibody incubation (Invitrogen, Karlsruhe, Germany, 1:25,000). Protein bands were visualised using SuperSignal© Western Blot Enhancer (Thermo Scientific, Karlsruhe, Germany, 46,640), SuperSignal© West Femto Maximum Sensitivity Substrate (Thermo Scientific, Karlsruhe, Germany, 34095), and Amersham Hyperfilm ECL (GE Healthcare, Freiburg, Germany, 28,906,839) films. Densitometric measurements were performed and analysed using the ImageJ Software, and β-actin was used as the reference protein. Statistical significance was assessed by an independent samples *t*-test. Data are presented as means ± SD of the values of three measurements in each group. Here, n = 3 independent samples. In all figures, *p*-values for statistical significance are as follows: * *p* < 0.05; ** *p* < 0.01; *** *p* < 0.001; **** *p* < 0.0001.

## Figures and Tables

**Figure 1 pharmaceuticals-16-00740-f001:**
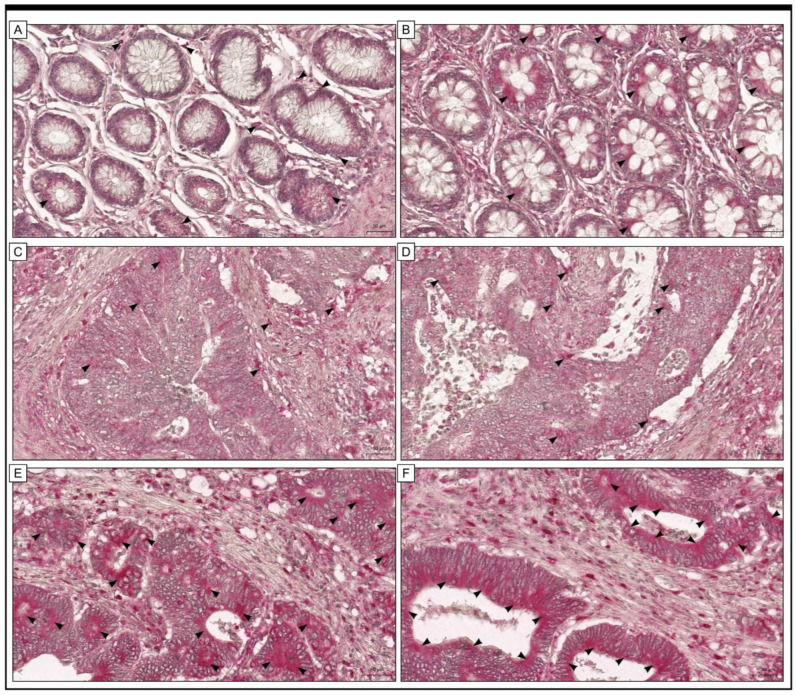
Representative microphotographs of immunohistochemical expression of Gpx-1 in colon adenocarcinoma tissue (**C**–**F**) and tissue margins with no cancerous lesions (adjacent non-tumour tissue) (**A**,**B**). (**A**,**B**) A low level of immunohistochemical reaction was detected in cells of non-pathological samples of the adjacent colon mucosa tissue margin. In samples of colon adenocarcinoma, the expression of Gpx-1 was described as low (**C**,**D**) or high (**E**,**F**). In some patients (**C**,**D**) Notch4 expression was described as low. The scale bar is 50 µm (**A**,**E**) and 100 µm (**B**,**C**,**D**,**F**).

**Figure 2 pharmaceuticals-16-00740-f002:**
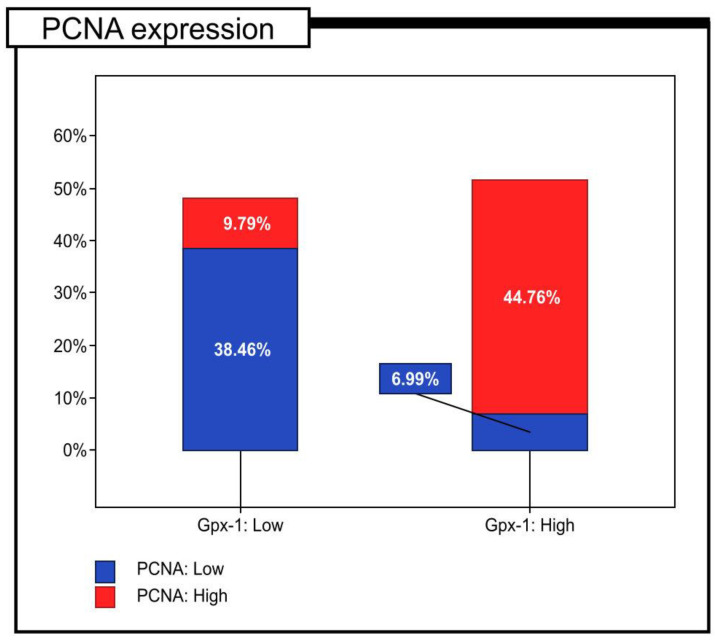
Percentage of immunohistochemical expression of PCNA defined as high and low expression in colon adenocarcinoma patients (*n* = 129).

**Figure 3 pharmaceuticals-16-00740-f003:**
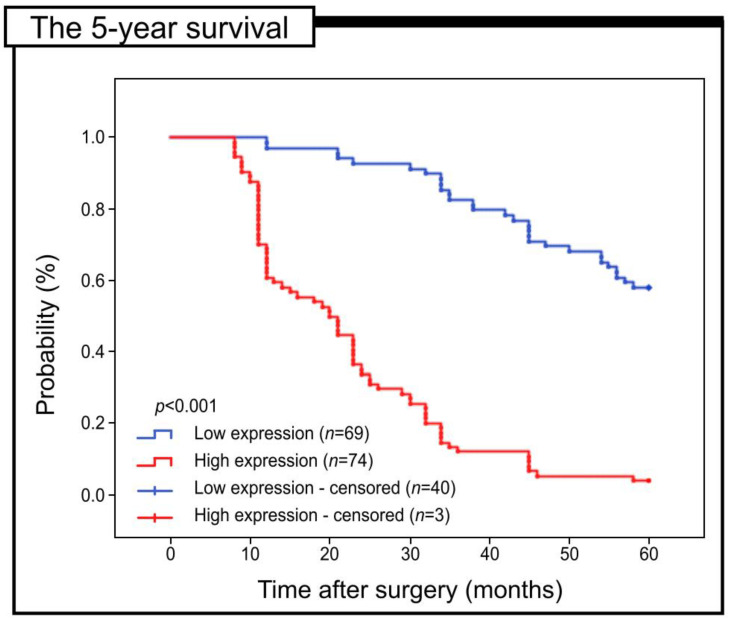
Kaplan–Meier curves of univariate analysis data (log-rank test) showing the 5-year survival rate for patients with high versus low levels of Gpx-1 immunohistochemical expression.

**Figure 4 pharmaceuticals-16-00740-f004:**
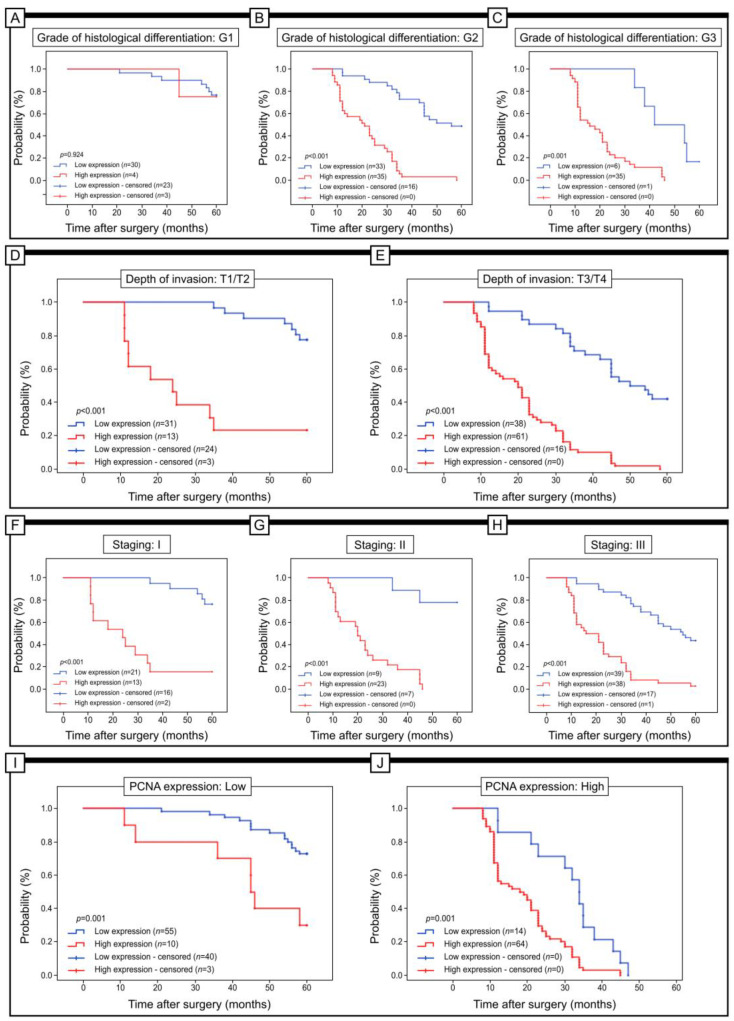
Kaplan–Meier curves of univariate analysis data (log-rank test) of patients with high versus low levels of Gpx-1 immunohistochemical expression. (**A**,**B**) Five-year survival of patients with G1 (**A**), G2 (**B**), and G3 (**C**) grade of differentiation; with T1/T2 (**D**) and T3/T4 depth of invasion (**E**); with staging I (**F**), staging II (**G**), and staging III (**H**); low level of immunohistochemical expression of PCNA (**I**) and high level of immunohistochemical expression of PCNA (**J**).

**Figure 5 pharmaceuticals-16-00740-f005:**
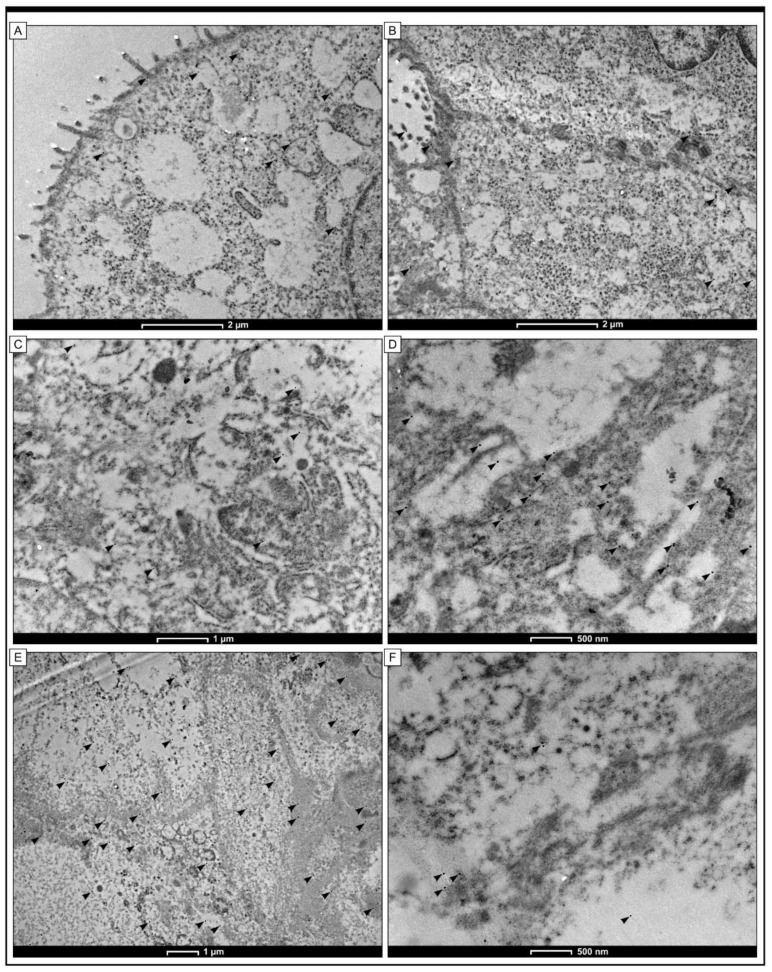
Immunogold labelling of Gpx-1 protein in cells within colon adenocarcinoma tissue. The small, black, and electron-dense granules (arrowheads) were detected in cells of the non-pathological colon tissue (**A**,**B**) and cells of colon adenocarcinoma samples (**C**–**F**). In cells of the non-pathological colon mucosa, a small number of electron-dense granules were detected within the cytoplasm of the apical part of the cells (**A**). In some cells, granules were also found in cisterns of the endoplasmic reticulum (**B**). In cancer cells, gold granules were visible to the mitochondria and cisterns of the endoplasmic reticulum. Moreover, the presence of Gpx-1 was detected in the plasma membrane and in the cytoplasm in close proximity to the plasma membrane (**E**). The scale bar is 1 µm (**C**,**E**), 2 µm (**A**,**B**), and 500 µm (**D**,**F**).

**Figure 6 pharmaceuticals-16-00740-f006:**
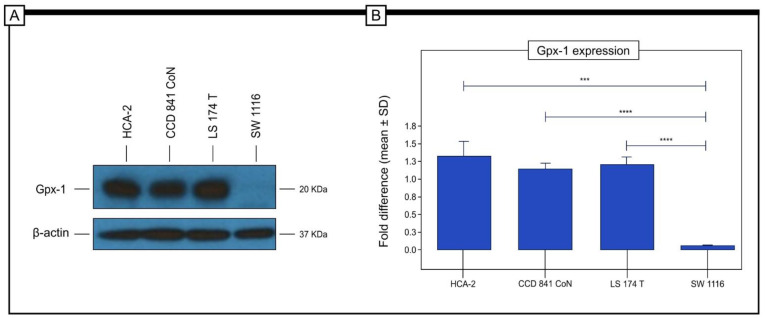
In vitro analysis of Gpx-1 expression in colorectal cancer cell lines representing a different type of colorectal cancer. The study revealed that the highest level of Gpx-1 protein expression was found in the HCA-2 cell line. The lowest level of expression was observed in the SW1116 cell line (**A**). Statistically significant differences in the expression of Gpx-1 protein were found between HCA-2 cells and SW1116 cells, between CCD 841 CoN cells and SW1116 cells, and between LS 174T cells and SW1116 cells. *p*-values of statistical significance are represented as follows: *** *p*  <  0.001; **** *p*  <  0.0001 (**B**).

**Table 1 pharmaceuticals-16-00740-t001:** Characteristics of patients included in the study (*n* = 143).

		*N* (Number of Cases)	%
Gender	Females	71	49.65
	Males	72	50.35
Age (years)	≤60 years	53	37.06
	61–75 years	50	34.97
	>75 years	40	27.97
	M ± SD	65.03 ± 12.72	
	Me (Q1–Q3)	65 (56–77)	
	Min–max	33–89	
Grade of histological differentiation (G)	G1	34	23.78
	G2	68	47.55
	G3	41	28.67
Depth of invasion (T)	T1	23	16.08
	T2	21	14.69
	T3	74	51.75
	T4	25	17.48
Regional lymph node involvement	N0	67	46.85
	N1	39	27.27
	N2	37	25.88
Location of tumour	Right-sided tumours	75	52.45
	Left-sided tumours	68	47.55
Angioinvasion	No	39	27.27
	Yes	104	72.73
Immunohistochemical expression of PCNA	Low	65	45.45
	High	78	54.55
Staging	I	34	23.78
	II	32	22.38
	III	77	53.84

**Table 2 pharmaceuticals-16-00740-t002:** Correlations between the expression of Gpx-1 protein and PCNA protein.

		The Immunoexpression Level of Gpx-1				*p*-Value
		Low		High		
PCNA expression	Low	55	(38.46%)	10	(6.99%)	*p* = 0.463
	High	14	(9.79%)	64	(44.76%)	*p* = 0.540

**Table 3 pharmaceuticals-16-00740-t003:** Correlations between the expression of Gpx-1 protein and clinicopathological characteristics in colon adenocarcinoma patients.

		The Immunoexpression Level of Gpx-1				*p*-Value
		Low		High		
Age (years)	≤60 years	29	(54.72%)	24	(45.28%)	*p* = 0.317
	61–75 years	20	(40.00%)	30	(60.00%)	
	>75 years	20	(50.00%)	20	(50.00%)	
Gender	Females	37	(52.11%)	34	(47.89%)	*p* = 0.359
	Males	32	(44.44%)	40	(55.56%)	
Grade of histological differentiation (G)	G1	30	(88.24%)	4	(11.76%)	*p* < 0.001
	G2	33	(48.53%)	35	(51.47%)	
	G3	6	(14.63%)	35	(85.37%)	
Depth of invasion (T)	T1	20	(86.96%)	3	(13.04%)	*p* < 0.001
	T2	11	(52.38%)	10	(47.62%)	
	T3	30	(40.54%)	44	(59.46%)	
	T4	8	(32.00%)	17	(68.00%)	
Regional lymph node involvement	N0	30	(44.78%)	37	(55.22%)	*p* = 0.737
	N1	20	(51.28%)	19	(48.72%)	
	N2	19	(51.35%)	18	(48.65%)	
Localisation	Left-sided tumours	39	(52.00%)	36	(48.00%)	*p* = 0.346
	Right-sided tumours	30	(44.12%)	38	(55.88%)	
Angioinvasion	Yes	34	(87.18%)	5	(12.82%)	*p* < 0.001
	No	35	(33.65%)	69	(66.35%)	
PCNA expression	Low	55	(84.62%)	10	(15.38%)	*p* < 0.001
	High	14	(17.95%)	64	(82.05%)	
Staging	I	21	(61.76%)	13	(38.24%)	*p* = 0.020
	II	9	(28.13%)	23	(71.88%)	
	III	39	(50.65%)	38	(49.35%)	

**Table 4 pharmaceuticals-16-00740-t004:** Univariate and multivariate analyses of various prognostic parameters in colon adenocarcinoma patients using Cox regression analyses.

Prognostic Parameter	Univariate Analysis			Multivariate Analysis		
	HR	95% CI	*p*-Value	HR	95% CI	*p*-Value
Gender	1.043	0.704–1.543	0.835	–	–	–
Age	1.008	0.993–1.023	0.316	–	–	–
Staging	1.370	1.072–1.752	0.012	0.971	0.573–1.645	0.913
Grade of histological differentiation	2.887	2.151–3.876	<0.001	1.221	0.804–1.854	0.348
Depth of invasion	1.908	1.509–2.412	<0.001	1.309	0.906–1.890	0.151
Regional lymph node involvement	1.332	1.054–1.682	0.016	1.034	0.691–1.547	0.870
Localisation	1.122	0.758–1.661	0.566	–	–	–
Immunohistochemical expression of Gpx-1 in cancer tissue	7.102	4.479–11.261	<0.001	2.755	1.554–4.883	0.001
Angioinvasion	3.940	2.227–6.970	<0.001	0.798	0.404–1.579	0.518
Expression of PCNA	20.226	10.602–38.586	<0.001	10.219	4.841–21.572	<0.001

## Data Availability

Data is contained within the article.

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
