# Peer review of "Immunohistochemical Expression of Glutathione Peroxidase 1 (Gpx-1) as an Independent Prognostic Factor in Colon Adenocarcinoma Patients"

_pharmaceuticals, 2023, doi:10.3390/ph16050740_

Round 1

Reviewer 1 Report

Brzozowa-Zasada et al, report an interesting topic of immunohistochemical expression of glutathione peroxidase 1 expression (Gpx-1) as an independent prognostic factor in colon adenocarcinoma patients. It is important findings but few things need to be clarified.

1.      In abstract part, please write “full name” at the first present in whole article. Eg. Gpx-1, TEM, PCNA, etc.

2.      In Introduction part, line 41-42, I do not find “mortality is expected to reach 60% by 2035” evidence in reference 4. Please make sure if you put a wrong reference.

3.      In Result and Methods parts, please mention how to define the high and low expressions of Gpx1 and PCNA. As you mention, Gpx1 expression in whole parts of cells, including cytoplasm, ER system, mitochondrial, nuclear membrane and nucleus. Does Gpx1 (+) cells mean high expression in whole cells or which part of cells? In pathological immunohistochemistry slides, you should can identify nuclear or cytoplasm or membrane stain.

4.      In 2.2., line 119 and Figure 2, high PCNA expression also shows high percentage of Gpx1 expression. Is PCNA expression in the same cells of Gpx1 expression ones? Co-immunohistochemistry stains can be performed.

5.      In 2.3. “he” prognostic significance………, please correct it.

6.      In Line 164-169, do you show the data in Figure 4? In you Figure 4I and J, no matter PCNA expression high or low patients, Gpx1 lower expression showed better overall survival (p=0.001, at both). What you write is a little bit confusion. Please mention the correlation between PCNA and Gpx1 expression.

7.      In 2.5 section, Discussion and methods parts, the cell lines name are different: WS1116 or SW1116; B41CoN or 841CoN?

8.      The in vitro cell lines study reports were different to clinical findings. In line 270~272, the reason is not convincible. In your methods part, two individual pathologists review pathological slides. It is not difficult to identify “tumor cells” from “microenvironment” by pathologists experiences. Some other experiments could be considered to find out the mechanism.

9. There are several abbreviations without full name at the first present in the manuscript. please correct them.

Reviewer 2 Report

The authors report an analysis of Polish patients with adenocarcinoma of the colon in which immunohistochemical staining for Peroxisase 1 (Gpx-1) expression was an independent prognostic factor.

The focus of the study is good, and the analysis is well done. Figures 1-3 and Table 3 are convincing with clinical data.

However, the results discussed in Figures 5-6 should be explored a little further.

At the very least, if there are high- and low-expressing cell lines (Fig. 6), then silencing or gene transfer should be performed for each, and at the very least, observations should be made, such as changes in growth kinetics.

Round 2

Reviewer 1 Report

It looks much better after revision.

Reviewer 2 Report

Now, authors modified their manuscript according to reviewer’s comments, adequately.